# Soil bacterial assemblage responses to wildfire in low elevation southern California habitats

**Elena Cox[1], André R. O. Cavalcanti[2], Edward J. Crane, III[2], Wallace M. Meyer, III**[2]*

**1** W. M. Keck Science Department, Claremont McKenna, Pitzer, Scripps Colleges, Claremont, CA, United States of America, **2** Department of Biology, Pomona College, Claremont, CA, United States of America

* wallace_meyer@pomona.edu

**Data Availability Statement:** Sequences from this study are available in GenBank's SRA database under Bioproject PRJNA699561 (2017 and 2019 samples) and Bioproject PRJNA398660 (2016 samples) A list of specific Biosamples is available

## Abstract

Understanding how wildfires and modification in plant assemblages interact to influence soil bacteria assemblages is a crucial step in understanding how these disturbances may influence ecosystem structure and function. Here, we resampled soil from three study sites previously surveyed in spring 2016 and 2017 and compared soil bacterial assemblages prior to and six months after (spring 2019) the 2018 Woolsey Fire in the Santa Monica Mountain National Recreation Area using Illumina sequencing of the 16S rRNA gene. All sites harbored both native California sage scrub and a non-native (grassland or forbland) habitat, allowing us to examine how fire influenced bacterial assemblages in common southern California habitats. Most results contrasted with our *a-priori* hypotheses: (1) richness and diversity increased following the fire, (2) heat/drought resistant and sensitive bacteria did not show consistent and differing patterns by increasing and decreasing, respectively, in relative abundance after the fire, and (3) bacterial assemblage structure was only minimally impacted by fire, with no differences being found between 2017 (pre-fire) and 2019 (post-fire) in three of the six habitats sampled. As sage scrub and non-native grasslands consistently harbored unique bacterial assemblages both before and following the fire, modifications in plant compositions will likely have legacy effects on these soils that persist even after a fire. Combined, our results demonstrate that bacterial assemblages in southern California habitats are minimally affected by fire. Because direct impacts of fire are limited, but indirect impacts, e.g., modifications in plant compositions, are significant, plant restoration efforts following a fire should strive to revegetate sage scrub areas to prevent legacy changes in bacterial composition.

## Introduction

Understanding how wildfires and non-native plants interact to influence soil microbial assemblages is crucial to understanding ecosystem structure and function. In the western United States, spread of invasive annuals and elevated human population densities increase ignition probabilities [1–3]. As invasive annuals are easy to ignite and are often the first to colonize an area after a fire, they reduce fire return intervals promoting their dominance over native plant

in the maps directory in the paper's GitHub repository: https://github.com/aroc110/Cox-et-al-2021. All the commands used in the qiime2 analyses are available in the paper's GitHub repository: https://github.com/aroc110/Cox-et-al-2021.

**Funding:** WMM, EJC, & AORC received funds from Howard Hughes Medical Institute, Chevy Chase, MD, through the Precollege and Undergraduate Science Education Program [grant 52007555]. EC received funds through Pitzer College, the Keck Science Department WMM, EJC, and AORC received funds from the Schenk Family, The funders had no role in study design, data collection and analysis, decision to publish, or preparation of the manuscript

**Competing interests:** The authors declare that they have no known competing financial interests or personal relationships that could have appeared to influence the work reported in this paper.

species, creating an invasive plant-fire cycle [4, 5]. While numerous studies have examined how fire influences above-ground plant assemblages [e.g., 6–9, highlighting only a few], fewer have examined how fires and modifications to plant assemblages interact to influence soil microbial assemblages. As soil microbes are key drivers of many important biogeochemical cycles [10–14], and can play a role in facilitating or inhibiting native plant recovery and plant invasions [15–17], understanding how these disturbances influence microbial assemblages can provide key insights into ecosystem structure and function.

In low-elevation areas of southern California, substantial portions of the endangered California sage scrub (hereafter, sage scrub) ecosystem are being type-converted to non-native grasslands, or forblands [18, 19]. Estimates suggest that 49% of the remaining sage scrub in southern California, not converted to urban/suburban habitats, have been type-converted to non-native grasslands since 1930, largely because of fire [7, 18]. Non-native mustard species (e.g., *Brassica nigra*, *Hirschfeldia incana*, and others) also colonize disturbed sage scrub areas forming monotypic stands [19–21]. While periodic fire is a natural aspect of the sage scrub ecosystem [22], invasion by non-native grass and forb species has increased fire frequencies and reduced fire-return intervals [23, 24]. Reduced fire-return intervals, drought, and elevated nitrogen deposition rates facilitate further invasion by non-native annuals, and reductions in native plant species [4, 8, 25]. As native sage scrub and non-native grasslands and forblands harbor different soil microbial assemblages [26–30], modifications in plant and microbial assemblages may alter many important ecosystem processes (e.g., C and N cycling and litter decomposition; [19, 27–29, 31–33]). In addition, soil microbes from soil collected under sage scrub and non-native plant communities can inhibit or facilitate plant germination and growth of con- and hetero-specific species, though effects are species and context dependent [17, 34–36]. Combined results demonstrate that modifications in plant and soil microbial community composition can impact native biodiversity, ecosystem function, and restoration efforts aimed at preserving the endangered sage scrub ecosystem.

While research on the effect of fire on soil microbial assemblages in low elevation areas in southern California remain scant, research examining how drought influences microbial composition and function may be informative for predicting microbial responses to fire. Immediately following a fire disturbance, soil is exposed to direct sunlight. Consequently, the soil is presumably drier and hotter, suggesting that recently burned areas may experience localized drought stress. For example, Adams et al. [37] hypothesized that lack of seasonal turnover in ants from summer to fall in a burned area was related to warm-dry conditions persisting later in the burned habitat relative to adjacent unburned sage scrub and non-native grasslands. Castro et al. [28] studied microbial communities in natural southern California habitats and found that interannual variation in precipitation was the leading contributor to microbial community fluctuations with microbial biomass, abundance, and activity reduced during drought years and in simulated drought treatments. Also, Castro et al. [28] found that drought-tolerant bacterial groups, Firmicutes and Actinobacteria had higher, and drought-sensitive bacteria, Proteobacteria, had lower relative abundances during drought. Similarly, Pérez-Valera et al. [38] found higher relative abundances of Firmicutes and Actinobacteria and lower relative abundances of Proteobacteria in soils following a fire in Spain.

While we have some insights on how fire may indirectly influence soil microbial assemblages, by modifying plant community structure and creating drought like conditions [27, 28], our understanding of how fire directly influences southern California soil microbial assemblages remains rudimentary [39]. Research from other Mediterranean-type ecosystems, with similar hot-dry summers and cool-moist winters, suggests that fire causes declines in microbial abundance and diversity (though phylogenic diversity can increase), and that succession favors drought-adapted groups [38, 40–42]. Research from California suggests that while decreases in

microbial activity, bacterial biomass, and diversity can follow a fire disturbance, fire effects are transient. For example, Gutknecht et al. [43] showed in a central California non-native grassland that extracellular enzyme activity after a wildfire was lower in the first and second year after a fire, but the effect was gone after 3 years. Similarly, Docherty et al. [44] found that belowground microorganisms were resistant to the effects of low severity fire in a central California non-native grassland, and that aboveground and nutrient effects were gone after 33 months. Fenn et al. [32] examined a fire-age gradient that contained sites that spanned 0 to 80 years since the last fire in the chaparral ecosystem, and found that microbial C did not differ across the fire-age-gradient, but that microbial processes differed under different plant species, again highlighting that indirect effects of fire (e.g., modifications to plant community structure) may have larger impacts on microbial structure and function than direct fire effects.

In this study, we opportunistically resampled soil from study sites originally surveyed by Caspi et al. [19, 27] and compared soil bacterial assemblages prior to and six months after the 2018 Woolsey Fire in the Santa Monica Mountains. Each site had native sage scrub and an adjacent non-native habitat type (either a non-native grassland or non-native forbland). We hypothesized that after a wildfire: (1) bacterial richness and diversity is reduced [39], (2) microbial assemblages are significantly modified, though we are unsure if differences between native sage scrub and non-native grasslands will remain, and (3) heat and drought tolerant bacterial groups, e.g., Firmicutes and Actinobacteria, have higher relative abundances, and drought sensitive bacteria, e.g., Proteobacteria, have lower relative abundances in both habitat types [28, 38]. We are not currently aware of any studies that have examined the direct impact of fire on bacterial assemblages in sage scrub and non-native habitats in low elevation areas of southern California. Additionally, few studies have looked at changes in microbial assemblages post-wildfire in Mediterranean-type ecosystems, as most utilize prescribed/experimental fires. This is likely because the limited and unpredictable opportunities to study natural wildfire events as opposed to prescribed fire [39]. By comparing soil bacterial assemblages in native sage scrub and non-native grass- or forb-lands before and after fire, we provide initial insights into how soil bacterial assemblages are impacted by wildfire in native and type-converted habitat types in southern California.

## Materials and methods

### Sample collection

Samples were collected from three sites in the Santa Monica Mountain National Recreation Area, CA, two close to the coast (~ 2 to 3 km from the ocean), Santa Monica Mountains (34.0341770, -118.8002590) and Zuma Ridge (34.0314740, -118.8120120), and one site further inland (~13 km from the coast), Cheseboro (34.1496260, -118.7351330). Each site contained native sage scrub habitat and an adjacent non-native (grass or forb) habitat. Soils at all three sites were broadly characterized as Mollisols [19, 27, 45]. Additional information on site and habitat characteristics and soil properties are described in Caspi et al. [19, 27]. Briefly, all sites contained sage scrub and an adjacent non-native habitat within 300 m. Sage scrub sites were characterized by dominance of native drought-deciduous shrubs (mostly *Artemisia californica* and *Salvia leucophylla*) with intermixed evergreen woody shrubs [46]. Sage scrub sites in the study were composed of < 10% invasive cover by visual estimation prior to the fire. The non-native habitat differed among the three sites. Non-native grasslands, found at Cheseboro and Zuma Ridge, were composed primarily of grasses, mostly *Bromus* spp. and *Avena* spp., while a non-native forbland, found at the Santa Monica Mountain site was composed of *Brassica* spp., *Sylibum mariabum*, and *Centaurea melitensis* and contained less than 5% native shrub cover by visual estimation.

In 2018, almost 100,000 acres of the Santa Monica Mountains burned in the Woolsey Fire [47] including vegetation in our three sites. Microbial assemblages from the Santa Monica Mountain site collected in spring 2016 were initially assessed by Caspi et al. [27]. Samples were not collected from Cheseboro or Zuma Ridge in 2016. While Caspi et al. [19] collected soil for assessment of C and N and bacterial and fungal concentrations from all three sites in spring 2017, they also collected additional soil samples for microbial assemblage assessments. These samples were placed in a cooler with dry ice after collection in the field and were placed into -80°C freezer within 6 hrs of collection. We resurveyed these three sites on 6-May-2019 (spring 2019) collecting samples using an identical approach. Samples collected in 2019 were also stored in a freezer (-80°C) prior to DNA extraction.

At each site and during each collection year, we collected 12 soil samples from the top 10 cm (A-horizon) of soil after removing litter (O-horizon), six from native sage scrub and six from non-native habitat type. We intentionally collected soil in the spring to limit confounding effects related to season.

In total, we analyzed 85 samples. The Santa Monica Mountain site was the only one for which we had samples for 2016, comprising 7 non-native forbland samples and 6 sage scrub samples. These were previously sequenced and analyzed by Caspi et al. [27] and provide an interesting comparison here to examine interannual fluctuations. For 2017 and 2019, we sequenced and analyzed 12 samples for each of the three sites, 6 sage scrub samples and 6 non-native vegetation samples. The 2017 samples were collected ~1.5 years prior to the fire and the 2019 samples were collected ~0.5 year after the fire.

## DNA extractions and sequencing

We used the exact same method to extract and sequence bacterial assemblages for the 2017 and 2019 samples as that employed by Caspi et al. [27] for the 2016 samples. We used Qiagen's DNeasy PowerSoil Kit to extract DNA according to the kit procedure without optional incubations. We used approximately 0.25 g soil from each sample and sent extracted DNA to Molecular Research LP (https://www.mrdnalab.com/) for sequencing using the 27F (AGAGTTTGAT CCTGGCTCAG) and 515R (TTACCGCGGCTGCTGGCAC) primers for bacterial 16S rRNA gene amplification [48].

Molecular Research LP (https://www.mrdnalab.com/) performed the sequencing of the samples using the following protocol: The HotStarTaq Plus Master Mix Kit (Qiagen, USA) was used for PCR (barcode on forward primer) with the following cycle: 3 minutes at 94°C, then by 30 cycles of 30 seconds at 94°C, 40 seconds at 53°C and 1 minute at 72°C, and followed by a final elongation step of 5 minutes at 72°C. PCR products were pooled and used to create an Illumina DNA library, then sequenced using Illumina MiSeq v3 2 × 300 bp sequencing according to manufacturer guideline (Illumina, San Diego, CA, USA).

The 2017 and 2019 samples we sequenced in this study are available in GenBank's SRA database under Bioproject PRJNA699561. The 2016 samples were previously sequenced by Caspi et al. [27] (Bioproject PRJNA398660). Note that this previous dataset contains samples from other sites, but only the samples from the Santa Monica Mountain site were used in this study. A list of all the Biosamples used in this study is in S1 Table, and scripts to download the data from GenBank are available in the paper's GitHub repository: https://github.com/aroc110/Cox-et-al-2021.

## Bioinformatics and analysis

Reads were processed using qiime2 [49]. Sequence reads were demultiplexed and adaptors and primers were removed using the cutadapt qiime plugin [50]. Because of little overlap between

the forward and reverse reads, only the forward reads were used in the analysis. Forward reads were processed using the dada2 qiime plugin [51] to generate a table of unique amplicon sequence variants (ASV) and their counts. ASVs were taxonomy classified using the vsearch option of qiime's feature-classifier plugin [52, 53] and the SILVA database [54, 55]. All the commands used in the qiime2 analyses are available in the paper's GitHub repository: https://github.com/aroc110/Cox-et-al-2021.

## Statistical analysis

To compare richness between habitats and years within each site, we used qiime2 to pool samples in each habitat for each year and to perform a rarefaction analysis comparing the number of amplicon sequence variants (ASVs) to the number of sequences in the pooled data. We then examined total ASV richness ($\gamma$ richness) using a permutation based analyses comparing number of ASVs to the number of sequences. Significance was determined when 95% confidence intervals did not overlap.

To test for differences in diversity (Shannon) between habitats and among years, and examine if heat and drought tolerant (e.g. Actinobacteria and Firmicutes) or sensitive (e.g. Proteobacteria) bacterial groups were more or less abundant following the fire, respectively, we ran twelve PERMANOVAs using the program PRIMER-E with the PERMANOVA+ add on [56]. First, we ran three PERMAVOVAs, one for each site, to test if diversity differed among habitats and years. For the three bacterial groups, we ran nine additional PERMA-NOVAs, one for each bacterial group from each site, to examine if average relative abundances differed between habitats (native and non-native) and years. Similarity matrices for these PERMANOVAs were created using Euclidian similarity index. Following significant PERMANOVA results for year for the Santa Monica Mountain site, we ran permutation t-tests to examine pair-wise differences between years ($\alpha$ = 0.017, 3 pairwise tests). Following significant results for habitat or the habitat x year interaction, we ran permutation-based t-tests comparing differences among habitats and years using data from each habitat and each year as contrasts. Bonferoni corrections were applied for the number of pairwise comparisons among sites.

As we expected differences among sites [27], we examined differences in bacterial assemblages among years and between habitats, both fixed factors, by running three separate two-factor PERMANOVAs, one for each site. An MDS plot was constructed for each site to visualize the influence of habitat and year (fire) on bacterial assemblages. Following significant tests for site, year, or site by year interactions, pairwise PERMANOVA tests were run to assess if bacterial assemblages differed among years within habitats and between habitats within years. We used a Bonferroni correction to correct for multiple testing. Corrected $\alpha$-values were 0.0071 for testing for differences among habitats within years (8 tests), and $\alpha$ = 0.005 for testing differences between years at each habitat type (10 tests). Similarity matrices for PERMANO-VAs examining differences in microbial assemblages were created using the relative abundance data of bacterial ASVs and Bray-Curtis distance.

## Results

### Species richness

Our findings contrast with our hypothesis that bacterial richness declines following a fire. We found higher $\gamma$ bacterial richness in the spring after the fire (2019) compared to the spring ~ 1.5 yr prior to the fire (2017) in both habitats at both Zuma Ridge and the Santa Monica Mountain site (Fig 1). Patterns for richness in 2016 at the Santa Monica Mountain site were mixed, with elevated richness in the non-native forbland and reduced richness in the sage

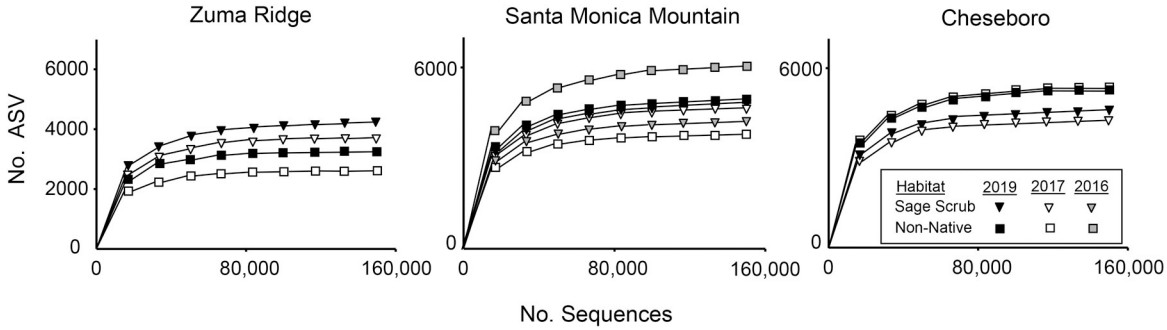

**Fig 1. Rarefaction curves comparing total number of ASVs to number of total sequences within each habitat and site across years (γ-diversity).** Error bars representing 95% confidence intervals are not shown, as the error was smaller than the symbols. As all habitat-year combinations differed from one another at each site, the relative position of the symbols demonstrates which habitat-year combination has higher richness.

scrub habitat. At Cheseboro, there was elevated richness in the sage scrub site in 2019 following the fire, but the opposite trend in the non-native grassland, the only instance we observed declining richness after the fire (Fig 1).

## Diversity

Diversity patterns were site dependent (Table 1, Fig 2), but there is no support for our hypothesis that diversity declines following a fire. At the Santa Monica Mountain site, diversity differed among years (Table 1), but diversity was higher in 2019 (post-fire) than in 2017 (pre-fire), and diversity did not differ between 2016 (~ 2.5 yr pre-fire) and 2019 (Fig 2). Diversity did not differ between years or habitat types at Zuma Ridge, but did differ between habitat types at Cheseboro (Table 1, Fig 2).

**Table 1. Results from PERMANOVA analyses examining differences in bacterial diversity and relative abundances of drought/heat tolerant and intolerant bacteria groups.**

| | Site | | | | | |
|---|---|---|---|---|---|---|
| *Metric* | Zuma Ridge | | Santa Monica Mtns. | | Cheseboro | |
| **Factor** | **Pseudo-F** | **P** | **Pseudo-F** | **P** | **Pseudo-F** | **P** |
| *Shannon Diversity* | | | | | | |
| **Year** | 0.48 | 0.495 | **5.30** | **0.006** | 1.47 | 0.246 |
| **Habitat** | 4.13 | 0.056 | 0.75 | 0.410 | **9.03** | **0.006** |
| **Year x Habitat** | 3.15 | 0.092 | 0.32 | 0.752 | 2.86 | 0.017 |
| *Actinobacteria Relative Abundance* (*drought tolerant*) | | | | | | |
| **Year** | 2.53 | 0.131 | 0.33 | 0.707 | 2.01 | 0.165 |
| **Habitat** | **25.69** | **< 0.001** | 0.85 | 0.362 | **5.83** | **0.026** |
| **Year x Habitat** | **11.895** | **0.003** | **5.71** | **0.005** | 0.17 | 0.681 |
| *Firmicutes Relative Abundance* (*drought tolerant*) | | | | | | |
| **Year** | **10.08** | **0.005** | 2.45 | 0.101 | **4.60** | **0.046** |
| **Habitat** | 2.13 | 0.165 | 0.77 | 0.395 | 3.66 | 0.067 |
| **Year x Habitat** | **7.32** | **0.016** | 1.11 | 0.357 | 2.03 | 0.175 |
| *Proteobacteria Relative Abundance* (*drought sensitive*) | | | | | | |
| **Year** | 0.75 | 0.401 | 1.57 | 0.222 | 3.22 | 0.083 |
| **Habitat** | **16.53** | **< 0.001** | **17.98** | **< 0.001** | 0.64 | 0.468 |
| **Year x Habitat** | 1.53 | 0.223 | **4.92** | **0.013** | 0.19 | 0.699 |

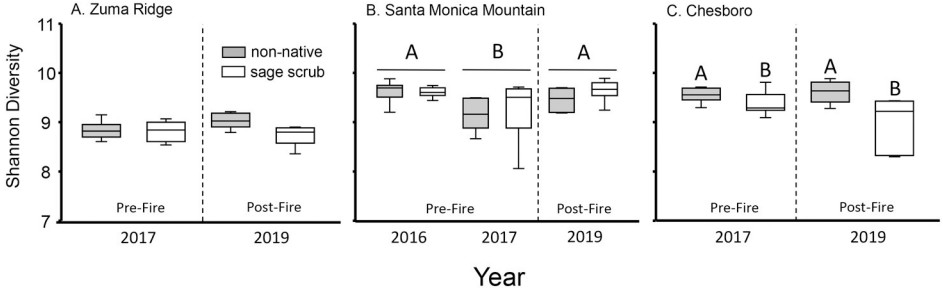

**Fig 2. Differences in Shannon diversity in native sage scrub and non-native habitats across sites and years.** Capital letters indicate differences in diversity among habitat types, while capital letters above horizontal lines represent differences in diversity among years within a site. There were no significant habitat by year interactions, so differences between habitats within and across years were not examined.

## Differences in relative abundance of heat/drought tolerant and sensitive bacteria

Patterns of relative abundance of Actinobacteria, a heat and drought tolerant bacterial taxa, did not match our prediction that the relative abundance of this group would increase following a fire. While abundance was elevated in the non-native habitat in the Santa Monica Mountain site following the fire in 2019 relative to 2017, abundance in 2019 were not different from those found in the non-native habitat in 2016. In contrast, Actinobacteria abundance was reduced in the non-native habitat at Zuma Ridge following the fire (Fig 3). No other differences in relative abundance of Actinobacteria before and after the fire in these habitats were observed (Table 1).

In contrast to Actinobateria, we found support for our hypothesis that Firmicutes, the other drought resistant bacterial group, increases in relative abundance following a fire. At Zuma Ridge we found elevated richness in the sage scrub habitat in 2019, and at Cheseboro, the relative abundance of Firmicutes was higher following the fire (Table 1, Fig 3). No differences in relative abundance of Firmicutes were observed among years within habitats at the Santa Monica Mountain site.

We predicted the relative abundance of Proteobacteria, the heat/drought sensitive group, would decrease following the fire. However, we found evidence of higher abundance of Proteobacteria in 2019 compared to 2017 at the Santa Monica Mountain site, though abundance did not differ between 2016 and 2019 (Table 1, Fig 3). While we did find differences in abundance among habitat types at Zuma Ridge, there were no differences between years at Zuma Ridge or between sites and years at Cheseboro (Table 1).

## Differences in bacteria assemblages between habitats and among years

At all three sites, bacterial assemblages differed among years (Zuma Ridge, Pseudo-F = 2.42, P = 0.002; Santa Monica Mountains, Pseudo-F = 3.12, P = < 0.001; Cheseboro, Pseudo-F = 1.69, P = 0.016) and habitats (Zuma Ridge, Pseudo-F = 12.24, P = < 0.001; Santa Monica Mountains, Pseudo-F = 12.12, P = < 0.001; Cheseboro, Pseudo-F = 2.98, P = < 0.001) and there were significant year by habitat interactions (Zuma Ridge, Pseudo-F = 2.42, P = 0.004; Santa Monica Mountains, Pseudo-F = 2.58, P = < 0.001; Cheseboro, Pseudo-F = 1.90, P = 0.006; Fig 4). While habitats consistently harbored unique bacterial assemblages (Table 2), pairwise comparisons comparing bacterial assemblages within habitats before and after the fire provided mixed results. At Zuma Ridge, bacterial assemblages differed between 2017 and 2019 in the non-native grassland habitat, but assemblages in the sage scrub habitat did not

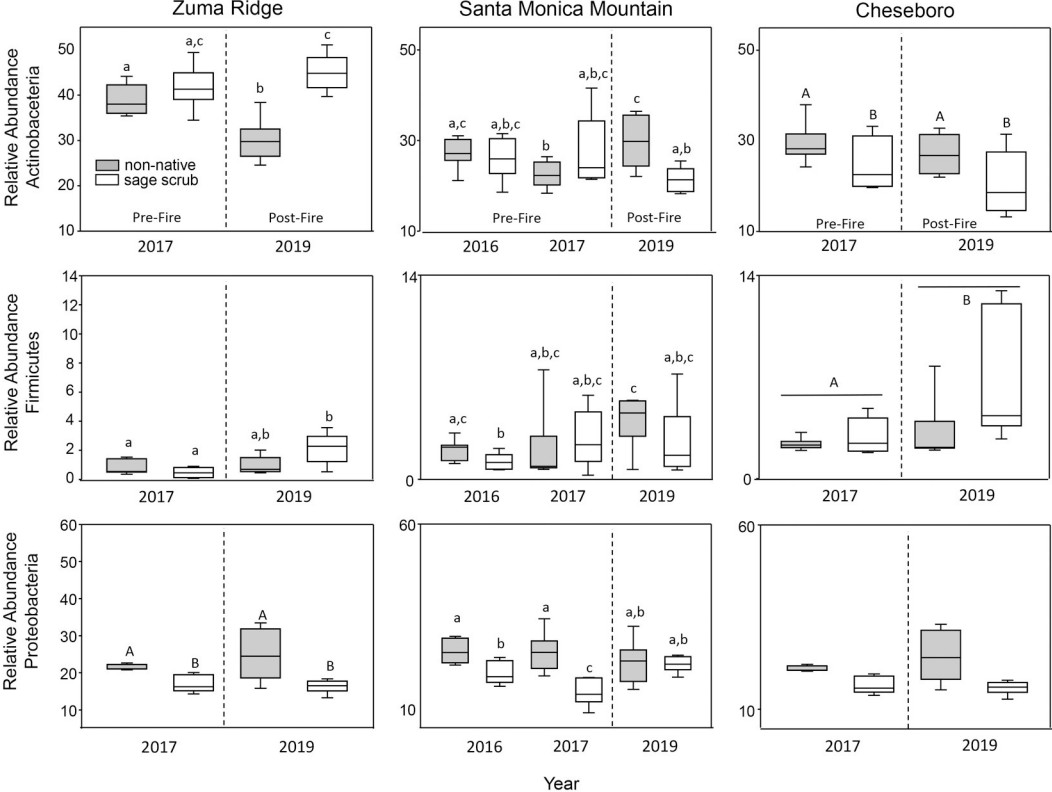

**Fig 3. Relative abundances of heat / drought tolerant bacteria *Actinobacteria*, *Firmicutes*, and drought sensitive bacteria *Proteobacteria* in sage scrub and non-native habitats at sampling sites: Zuma Ridge, Santa Monica Mountain, and Cheseboro in 2016, 2017, and 2019.** Capital letters indicate differences in diversity among habitat types, while capital letters above horizontal lines represent differences in diversity among years within a site. Lower case letters indicate differences between habitats within and across years.

differ (Table 3, Fig 4). Comparisons among 2016, 2017 and 2019 at the Santa Monica Mountain site reveled that while bacterial assemblages in sage scrub in 2017 differed from those in sage scrub in 2019, sage scrub bacterial assemblages in 2016 and 2019 did not differ (Table 3, Fig 4). At Cheseboro, bacterial assemblages in both habitats did not differ before and after the fire (Table 3).

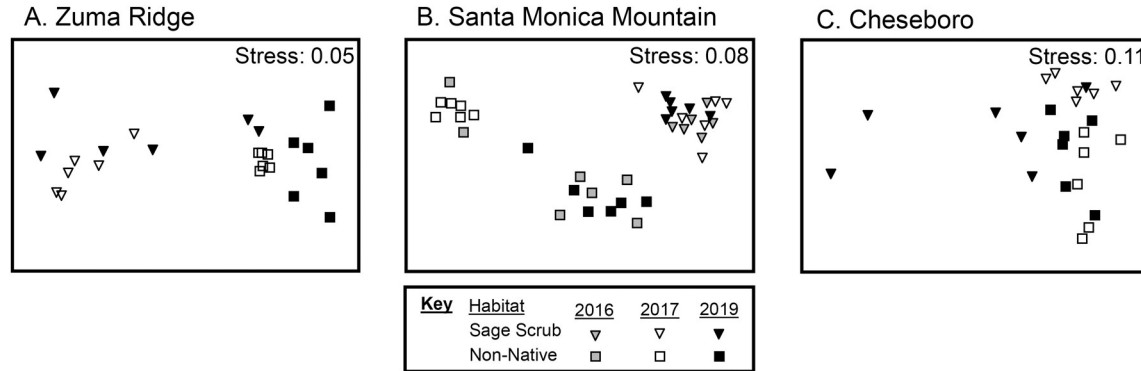

**Fig 4. MDS ordination plots of sites: Zuma Ridge, Santa Monica Mountains, and Cheseboro in 2016, 2017, and 2019 according to the relative abundance and composition of bacterial ASVs at each site.**

**Table 2. Comparisons between bacterial assemblages in native sage scrub and non-native habitats within years.**

| | Site | | | | | |
|---|---|---|---|---|---|---|
| | Zuma Ridge. | | Santa Monica Mtns | | Cheseboro | |
| Year | Pseudo-F | P | Pseudo-F | P | Pseudo-F | P |
| **2016** | | | 2.112 | 0.002* | | |
| **2017** | 3.561 | 0.002* | 2.998 | 0.002* | 1.675 | 0.002* |
| **2019** | 2.072 | 0.003* | 2.113 | 0.002* | 1.464 | 0.003* |

Asterisks indicate significant p-values below adjusted α-values (α = 0.007 with Bonferroni correction) to correct for multiple testing.

## Discussion

Our results suggest that direct fire impacts on soil bacterial assemblages in low-elevation southern California systems are complex, occasionally counterintuitive, and can be difficult to assign to the fire disturbance. For example, our results oppose those of Pressler et al. [39], whose meta-analysis found that richness and diversity often decline following a fire. Our study found that richness and diversity were either not impacted or elevated following a fire. For example, γ richness was higher within all habitats at each site, except for the non-native grassland at Cheseboro, in spring 2019, the spring immediately after the fire, compared to richness in the same sites in spring 2017, approximately 1.5 years prior to the fire. Pérez-Valera et al. [42] found that phylogenetic diversity increased after a fire before competitively dominant microbes recovered. While further study is required, increases in richness and diversity immediately following a fire in our study support the hypothesis that recently burned areas may initially allow for colonization of diverse bacteria taxa. However, our samples in 2019 were collected in the spring immediately following the fire, which is before the extreme hot-dry environmental conditions of summer, which are hypothesized to be a strong selecting force for microbial assemblages in Mediterranean semi-arid environments [57]. Consequently, the timing of sample collection may significantly influence diversity patterns and interpretations, especially in southern California systems. Similar to patterns observed here for soil bacteria, many ground-dwelling invertebrate taxa are also not directly impacted by fire in California and in other semi-arid environments [37, 58–62], highlighting that, while often assumed, fires do not necessarily lead to declines in soil richness and diversity.

**Table 3. Comparisons between years within native sage scrub and non-native habitats (non-native forbland in the Santa Monica Mountain site, and non-native grasslands in Zuma Ridge and Cheseboro).**

| | Site | | | | | |
|---|---|---|---|---|---|---|
| Year | Zuma Ridge | | Santa Monica Mtns. | | Cheseboro | |
| Habitat | Pseudo-F | P | Pseudo -F | P | Pseudo -F | P |
| **2016–2017** | | | | | | |
| Sage Scrub | | | 1.307 | 0.007 | | |
| Non-native | | | 2.053 | 0.004* | | |
| **2016–2019** | | | | | | |
| Sage Scrub | | | 1.320 | 0.016 | | |
| Non-native | | | 1.269 | 0.053 | | |
| **2017–2019** | | | | | | |
| Sage Scrub | 1.390 | 0.062 | 1.369 | 0.002* | 1.531 | 0.006 |
| Non-native | 1.780 | 0.003* | 2.543 | 0.002* | 1.086 | 0.226 |

Asterisks indicate significant p-values below adjusted α-values (α = 0.005 with Bonferroni correction) to correct for multiple testing.

Moreover, it is difficult to determine to what extent fire directly impacts soil microbial assemblages. While we found that bacterial assemblages did differ between 2017 (pre-fire) and 2019 (post-fire) in both habitats at the Santa Monica Mountain site and in the non-native grassland at Zuma Ridge, bacterial assemblages did not differ in either habitat at Cheseboro or within sage scrub at Zuma Ridge. Interestingly, bacterial assemblages in native sage scrub at the Santa Monica Mountain site did not differ between 2016, 2.5 years prior to the fire, and 2019, highlighting that changes in microbial assemblages after the fire are consistent with inter-annual fluctuations. Malik et al. [29] found that when stressed, grass litter assemblages allocate resources to survival relative to growth, which may explain mechanisms that allow bacterial assemblages to remain intact after a fire disturbance. These mechanisms may also help explain why bacterial assemblages in our most inland site, Cheseboro, did not differ pre- or post-fire in either habitat, as bacteria in warmer inland sites may be preconditioned for stress. Though more research is required, results could suggest that bacterial assemblages in inland sites may be more resilient to fire than those on the coast. In retrospect, we are not surprised that bacterial assemblages in Mediterranean systems that evolved with fire and extreme seasonal and interannual drought would be resistant to a fire disturbance. Though studies in other areas of California and other Mediterranean systems have shown that bacterial communities usually take multiple years (~ 3) to recover from fire [40, 41, 44, 57], our study suggests that direct effects of fire on soil bacterial assemblages are minimal, and that assemblages may not differ from those before the fire.

In contrast to the direct effects of fire, indirect effects of fire, i.e., changing plant community composition, would likely have significant and long-lasting, "legacy" effects on soil microbial assemblages. Picket et al [16] demonstrated that soils conditioned by non-native species can have "legacy" effects on soil microbial communities that last decades and persist after restoration efforts. In our study, sage scrub and non-native grasslands consistently harbored unique bacterial assemblages both before and after a fire highlighting that changes associated with differing plant compositions persist. Since differences in bacterial assemblages between habitat types did not change following a fire, results may suggest that previous research on plant-soil feedbacks may provide key insights for restoration efforts, specifically how soil types may facilitate native and non-native plant recruitment following a fire [34, 35]. In addition, sage scrub soils have lower respiration and decomposition rates and store more carbon than those in non-native grasslands, though non-native forblands with mustards may store more carbon than sage scrub soils, suggesting that ecosystem function will likely be more impacted by changes in plant composition, than direct impacts of fire on microbes [19, 27, 31, 33, 63].

Though we were hopeful that similar patterns reported by Pérez-Valera et al. (2019) in Spain after a fire, and Castro et al. [28] in southern California, after a drought, would lead to a predictive framework for understanding how fire influences bacterial assemblages in Mediterranean ecosystems, we found little support for the hypothesis that the relative abundance of heat/drought resistant bacterial phyla (Actinobacteria) increase and relative abundance of drought/heat sensitive bacteria (Proteobacteria) decrease immediately following a fire. Contrary to our hypothesis, we found that the relative abundances of Actinobacteria were unaffected by fire or declined following the fire at Zuma Ridge and Cheseboro. At the Santa Monica site, while we did find elevated abundances post-fire in 2019, relative to 2017, relative abundances in 2016 and 2019 did not differ, again suggesting that fluctuations are consistent with interannual fluctuations and difficult to assign to the fire disturbance. These results also partially support the assertion that drought, more than fire, may be the most significant disturbance influencing soil microbial assemblages [28, 30]. Rainfall in Los Angeles in 2015–2016 (which encompasses spring 2016) was 12.9 cm below average, while rainfall in 2016–2017 and 2018–2019 was 10.8 and 10.4 cm above average [64]. Despite fluctuations in annual

precipitation, the relative abundance of Proteobacteria did not differ among years at any site, though differences were observed among habitats. Patterns for Firmicutes relative abundances did support our hypotheses that this drought tolerant group of bacteria may increase following a fire, as we saw increases in relative abundances in 2019 (post-fire) relative to 2017 (pre-fire) abundances at Cheseboro and in the sage scrub habitat at Zuma Ridge, though no patterns were observed across years at the Santa Monica Mountain site. As most patterns regarding the relative abundances of both drought/heat tolerant and sensitive groups contrast with our *a-priori* hypotheses, results suggest that we remain at our infancy for developing a predictive framework for how fire impacts bacterial assemblages, though these patterns may further support the idea that fire has minimal direct impacts in soil bacterial assemblages. Therefore, drought might have stronger impacts on these assemblages than fire, and using patterns associated with drought to infer patterns that may be associated with fire may not be appropriate.

Studying the effects of wildfire are logistically difficult. In this study, we had a unique opportunity as a wildfire burned three of our previous study sites, giving us before and after data at the same locations. We recognize that our experiment did not have control sites, as the three sites that were previously surveyed, all burned, making comparisons between areas that did not burn across years impossible. Though many other studies examining soil microbial responses to fire use unburned reference sites [e.g., 40], we argue this approach can introduce more variability due to high levels of variability in bacterial assemblages across sites, especially if unburned reference sites are far from burned sites as bacterial assemblages differ across short distances [65]. In our study system, even sites that are relatively close to one another (less than a few km separate Zuma Ridge from the Santa Monica Mountain site) harbor unique bacterial assemblages. Consequently, determining if reference sites outside the fire perimeter differ because of the fire or because of differences in micro-geographic location would be extremely difficult. Therefore, we argue that having microbial data from the exact same sampling locations before and after a wildfire provide the best opportunity to understand the impacts of fire on these assemblages, particularly if you have data from multiple years prior to the fire to enable examination of interannual fluctuations, as we did for the Santa Monica Mountain site.

Our results indicate that while direct effects of fire in low elevation southern California systems are minimal, indirect effects, changes in plant community structure resulting from the disturbance, could have significant long-lasting legacy impacts on soil bacterial assemblage structure. Contrary to our hypotheses, richness and diversity increased following the fire highlighting that fires do not lead to declines in bacterial ASV richness and diversity. While increased richness and diversity following a fire suggests that fires may facilitate bacterial recruitment, these changes only minimally impacted bacterial assemblage structure, with no differences being found between 2017 (pre-fire) and 2019 (post-fire) samples in three of the six habitats sampled. As sage scrub and non-native grasslands consistently harbor unique bacterial assemblages both before [26–30] and following a fire (this study), knowing plant compositions prior to the fire may provide key insights for restoration efforts and predicting ecosystem functioning. For example, because microbial assemblages in sage scrub and non-native grasslands and forbland soils have differing effects on plant germination and growth [16–17, 34–36], knowing plant compositions prior to the fire could help predict plant succession patterns and inform restoration efforts. Similarly, as key ecosystem processes differ in sage scrub and non-native grass and forbland soils, knowing plant compositions prior to the fire may help predict how an ecosystem may function after a fire [19, 27, 31, 33, 63]. Our hypotheses that heat/drought resistant and sensitive bacteria would show differing patterns by increasing and decreasing, respectively, in relative abundance after the fire, was not observed and is consistent with minimal change in the assemblage structure. However, future studies should consider

sampling microbial communities in both the spring and summer, or cool-moist and hot-dry Mediterranean seasons. following a fire to examine the direct impacts of fire (spring samples) and the indirect fire effects of enhanced drought-like conditions (summer samples). Combined, our results demonstrate that bacteria assemblages in the Santa Monica Mountains National Recreational area are little affected by fires. Further studies are required to elucidate if this trend is applicable to other semi-arid systems and in places where fire is common. Because direct impacts of fire are limited, but indirect impacts, e.g., modifications in plant compositions, are significant, plant restoration efforts following a fire should strive to revegetate sage scrub areas to prevent legacy changes in microbial composition.

## Supporting information

**S1 Table. Accession numbers of all the Biosamples used in this study.**
(XLSX)

## Acknowledgments

We thank former Claremont College undergraduate students Tal Caspi, Lauren Hartz, Alondra Soto Villa, Jenna Loesberg, Leo Estrada, Anna V. Dowling, Erin Su, and Maxim Leshchinskiy for their support with collecting and processing samples. We thank the Santa Monica Mountains National Recreation Area [permit SAMO-00160] for access to our study sites.

## Author Contributions

**Conceptualization:** Elena Cox, Wallace M. Meyer, III.

**Data curation:** Elena Cox, André R. O. Cavalcanti.

**Formal analysis:** Elena Cox, André R. O. Cavalcanti, Edward J. Crane, III, Wallace M. Meyer, III.

**Funding acquisition:** Elena Cox, Edward J. Crane, III, Wallace M. Meyer, III.

**Investigation:** Elena Cox, Edward J. Crane, III.

**Methodology:** Elena Cox, André R. O. Cavalcanti, Wallace M. Meyer, III.

**Project administration:** Wallace M. Meyer, III.

**Resources:** André R. O. Cavalcanti, Wallace M. Meyer, III.

**Software:** André R. O. Cavalcanti, Wallace M. Meyer, III.

**Supervision:** Wallace M. Meyer, III.

**Validation:** Elena Cox, André R. O. Cavalcanti.

**Visualization:** Elena Cox, André R. O. Cavalcanti, Wallace M. Meyer, III.

**Writing – original draft:** Elena Cox.

**Writing – review & editing:** Elena Cox, André R. O. Cavalcanti, Edward J. Crane, III, Wallace M. Meyer, III.

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
