## [Decision Letter · Decision Letter 0]

10 Feb 2022

PONE-D-21-38710Soil bacterial assemblage responses to wildfire in low elevation southern California habitatsPLOS ONE

Dear Dr. Meyer

Thank you for submitting your manuscript to PLOS ONE. After careful consideration, we feel that it has merit but does not fully meet PLOS ONE’s publication criteria as it currently stands. Therefore, we invite you to submit a revised version of the manuscript that addresses the points raised during the review process. 

Please submit your revised manuscript by Mar 27 2022 11:59PM, If you will need more time than this to complete your revisions, please reply to this message or contact the journal office at plosone@plos.org. Please include the following items when submitting your revised manuscript:A rebuttal letter that responds to each point raised by the academic editor and reviewer(s). You should upload this letter as a separate file labeled 'Response to Reviewers'.A marked-up copy of your manuscript that highlights changes made to the original version. You should upload this as a separate file labeled 'Revised Manuscript with Track Changes'.An unmarked version of your revised paper without tracked changes. You should upload this as a separate file labeled 'Manuscript'.

We look forward to receiving your revised manuscript.

Kind regards,

Tunira Bhadauria, Ph.D.

Academic Editor

PLOS ONE

Journal Requirements:

“WMM, EJC, & AORC received funds from Howard Hughes Medical Institute, Chevy Chase, MD, through the Precollege and Undergraduate Science Education Program [grant 52007555].

EC received funds through Pitzer College, the Keck Science Department

WMM, EJC, and AORC received funds from the Schenk Family,”

“This work was supported by the Pomona College Biology Department, the Howard Hughes Medical Institute, Chevy Chase, MD, through the Precollege and Undergraduate Science Education Program [grant 52007555], Pitzer College, the Keck Science Department, and funding from the Schenk family.”

“WMM, EJC, & AORC received funds from Howard Hughes Medical Institute, Chevy Chase, MD, through the Precollege and Undergraduate Science Education Program [grant 52007555].

EC received funds through Pitzer College, the Keck Science Department

WMM, EJC, and AORC received funds from the Schenk Family,”

4. We noted in your submission details that a portion of your manuscript may have been presented or published elsewhere. [DETAILS AS NEEDED] Please clarify whether this [conference proceeding or publication] was peer-reviewed and formally published. If this work was previously peer-reviewed and published, in the cover letter please provide the reason that this work does not constitute dual publication and should be included in the current manuscript.

Reviewers' comments:

Reviewer's Responses to Questions

**Comments to the Author**

1. Is the manuscript technically sound, and do the data support the conclusions?

Reviewer #1: Yes

Reviewer #2: Yes

2. Has the statistical analysis been performed appropriately and rigorously? 

Reviewer #1: Yes

Reviewer #2: Yes

3. Have the authors made all data underlying the findings in their manuscript fully available?

Reviewer #1: Yes

Reviewer #2: Yes

4. Is the manuscript presented in an intelligible fashion and written in standard English?

Reviewer #1: Yes

Reviewer #2: Yes

5. Review Comments to the Author

Reviewer #1: Manuscript recommended for publication

1Manuscript reserch results indicate that while direct effects of fire in low elevation southern California

systems are minimal, indirect effects, changes in plant community structure resulting from the

disturbance, could have significant long-lasting legacy impacts on soil bacterial assemblage

structure are remarkable

2.Richness and diversity increased following the fire highlighting that fires do not lead to declines in bacterial ASV richness and diversity. While20 increased richness and diversity following a fire suggests that fires may facilitate bacterial recruitment, these changes only minimally impacted bacterial assemblage structure, with no differences being found between 2017 (pre-fire) and 2019 (post-fire) samples in three of the six habitats sampled.These findings are remarkable

3.Please explain in about DNa isolation method incase you did any modification in refered procedure?

4.Note that Bioproject PRJNA398660 contains samples from other sites, only the bacterial samples from the Santa Monica Mountains site were used in this study. A list of these specific Biosamples is available in the maps directory in the paper’s GitHub repository: https://github.com/aroc110/Cox-et-al-2021.Pl improve this centence

5.How can fire affect bacterial and biological diversity pl clarify in discussion

6.Pl explain about future prospective also

7.Pl elaborate methodology

Reviewer #2: Dear Author

The MS is of high quality and technically sound. However various questions arise after following the study parameters.

1. The richness and diversity of bacteria are dependent on both biotic and abiotic factors. In the present study, only fire has been taken to see the effect on bacteria community. How this study can neglect the effect of other parameters while studying in the open ecosystem.

2. As per your hypothesis, wildfire should negatively affect the microbial diversity but your findings suggest that the effect was positive or negligible. What are the chances that these effects are not because of fire but may be due to microbial interaction with other abiotic factors like edaphic factors?

3. The study does not have control, therefore the results do not seems reliable or you can say it may have other influences too.

4. It has been mentioned that diversity was higher in 2019 (post-fire) than 2017 (Pre-fire) but the diversity did not differ between 2016 and 2019. It means in 2017, diversity had fluctuated. Any reason?

5. Fire usually affects various soil properties which ultimately affects the bacterial community. In this study, soil parameters have not been studied at concerned site.

6. Do the studied sites have frequent wildfire history? The bacterial populations of the concerned sites may have resistant bacteria whose diversity has been modulated by wildfire in the past history before your study.

7. With the sequencing data, what could be elaborated is not reflected in result and discussion portion.

8. MS is not written in crispy fashion. Too large introduction, material methods and discussion.

9. Sometimes sentences are too large to understand like line number 81 to 85.

10. Rectify the line number 169, 30-36 30s cycles at 940 C?

11. The application of the study has not been reflected in abstract and discussion portion. Please look into that.

12. Check the references as per journal guidelines.

6. PLOS authors have the option to publish the peer review history of their article (what does this mean?). If published, this will include your full peer review and any attached files.

Reviewer #1: No

Reviewer #2: No

---

## [Author Response · Author response to Decision Letter 0]

2 Mar 2022

Please refer to attached "Response to Reviewers" document where we address each of the reviewers requests/recommendations.

---

## [Editor Report · Decision Letter 1]

18 Mar 2022

Soil bacterial assemblage responses to wildfire in low elevation southern California habitats

PONE-D-21-38710R1

Dear Dr. Meyer

We’re pleased to inform you that your manuscript has been judged scientifically suitable for publication and will be formally accepted for publication once it meets all outstanding technical requirements.

Kind regards,

Tuneera Bhadauria, Ph.D.

Academic Editor

PLOS ONE

Additional Editor Comments (optional):

After reading the revised manuscript, I am pleased to note that the authors have adequately responded to all of the recommendations made by both reviewers. They've also included the changes/suggestions in the text. I believe the manuscript has been sufficiently updated to make it worthy of publication in the journal, and hence recommend that it be accepted for publication.
---

## [Editor Report · Acceptance letter]

30 Mar 2022

PONE-D-21-38710R1 

Soil bacterial assemblage responses to wildfire in low elevation southern California habitats 

Dear Dr. Meyer III:

I'm pleased to inform you that your manuscript has been deemed suitable for publication in PLOS ONE. Congratulations! Your manuscript is now with our production department. 

Kind regards, 

on behalf of

Dr. Tunira Bhadauria 

Academic Editor

PLOS ONE